# Biochar-Mediated Control of Metabolites and Other Physiological Responses in Water-Stressed *Leptocohloa fusca*

**DOI:** 10.3390/metabo13040511

**Published:** 2023-04-01

**Authors:** Khansa Saleem, Muhammad Ahsan Asghar, Ali Raza, Hafiz Hassan Javed, Taimoor Hassan Farooq, Muhammad Arslan Ahmad, Altafur Rahman, Abd Ullah, Baiquan Song, Junbo Du, Fei Xu, Aamir Riaz, Jean W. H. Yong

**Affiliations:** 1Department of Horticultural Sciences, The Islamia University of Bahawalpur, Bahawalpur 63100, Pakistan; 2Department of Biological Resources, Agricultural Institute, Centre for Agricultural Research, ELKH, 2 Brunzvik St., 2462 Martonvásár, Hungary; 3Chengdu Institute of Biology, Chinese Academy of Sciences, University of Chinese Academy of Sciences, Beijing 100049, China; 4College of Agronomy, Sichuan Agricultural University, Wenjiang, Chengdu 611130, China; 5Bangor College China, A Joint Unit of Bangor University and Central South University of Forestry and Technology, Changsha 410004, China; 6Shenzhen Key Laboratory of Marine Bioresource and Eco-Environmental Science, College of Life Sciences and Oceanography, Shenzhen University, Shenzhen 518060, China; 7Xinjiang Key Laboratory of Desert Plant Roots Ecology and Vegetation Restoration, Xinjiang Institute of Ecology and Geography, Chinese Academy of Sciences, Urumqi 830011, China; 8Engineering Research Center of Agricultural Microbiology Technology, Ministry of Education & Heilongjiang Provincial Key Laboratory of Ecological Restoration and Resource Utilization for Cold Region & School of Life Sciences, Heilongjiang University, Harbin 150080, China; 9Applied Biotechnology Center, Wuhan University of Bioengineering, Wuhan 430415, China; 10Department of Biosystems and Technology, Swedish University of Agricultural Sciences, 23456 Alnarp, Sweden

**Keywords:** biochar, metabolites, drought, antioxidants, photosynthesis

## Abstract

We investigated biochar-induced drought tolerance in *Leptocohloa fusca* (Kallar grass) by exploring the plant defense system at physiological level. *L. fusca* plants were exposed to drought stress (100%, 70%, and 30% field capacity), and biochar (BC), as an organic soil amendment was applied in two concentrations (15 and 30 mg kg^−1^ soil) to induce drought tolerance. Our results demonstrated that drought restricted the growth of *L. fusca* by inhibiting shoot and root (fresh and dry) weight, total chlorophyll content and photosynthetic rate. Under drought stress, the uptake of essential nutrients was also limited due to lower water supply, which ultimately affected metabolites including amino and organic acids, and soluble sugars. In addition, drought stress induced oxidative stress, which is evidenced by the higher production of reactive oxygen species (ROS) including hydrogen peroxide (H_2_O_2_), superoxide ion (O_2_^−^), hydroxyl ion (OH^−^), and malondialdehyde (MDA). The current study revealed that stress-induced oxidative injury is not a linear path, since the excessive production of lipid peroxidation led to the accumulation of methylglyoxal (MG), a member of reactive carbonyl species (RCS), which ultimately caused cell injury. As a consequence of oxidative-stress induction, the ascorbate–glutathione (AsA–GSH) pathway, followed by a series of reactions, was activated by the plants to reduce ROS-induced oxidative damage. Furthermore, biochar considerably improved plant growth and development by mediating metabolites and soil physio-chemical status.

## 1. Introduction

Due to the severity of abiotic challenges such as drought, salinity, heavy metals, severe temperatures, and radiation, the natural environment is being maltreated. These environmental stresses substantially reduce plant productivity and development in many ways [1]. Drought is one of the abiotic stressors that has deleterious effects on plant developmental processes [2,3]. Under abiotic stress, plants undergo a series of changes from morpho-physiological to biochemical alterations. Plant cellular and molecular studies revealed that reactive oxygen species (ROS) such as hydroxyl ion (OH^−^), singlet oxygen (^1^O_2_), superoxide ion (O_2_^−^), and hydrogen peroxide (H_2_O_2_) are the primary and inescapable outcome of main aerobic reactions [4]. Previous evidence found that these reactive species are both toxic and signaling in nature [5,6]. However, their toxic and signaling effects entirely depend on the plant varieties, soil type, temperature, and water availability. In a very small amount, these species work as signaling molecules in response to abiotic stress, while in excessive amounts, these species induce damage to membranes, amino and organic acids, proteins, and DNA/RNA which leads to programmed cell death in plants [7,8,9]. Earlier, it was considered that ROS alone are responsible for the oxidative damage in plants. However, some recent studies revealed that excessive ROS production increases lipid peroxidation-derived molecules such as ketones, aldehydes, malondialdehyde (MDA), and methylglyoxal (MG) [10,11]. These highly unstable reactive molecules cause irreversible damage to plants which ultimately leads to cell death [5,12]. These reactive molecules, even in small amounts, have the ability to alter the biological and metabolic systems in plants [11].

As a consequence of drought-induced oxidative stress, plants activate their cellular defense mechanism [13]. This defense system consist of enzymatic and non-enzymatic antioxidants including superoxide dismutase (SOD), catalase (CAT), peroxidase (POD), glutathione peroxidase (GPX), glutathione reductase (GR), glutathione S-transferase (GST), ascorbate (AsA), glutathione (GSH), monodehydroascorbate reductase (MDHAR), dehydroascorbate reductase (DHAR), and dehydroascorbate (DHA) [14]. These enzymes are essential for the detoxification and scavenging of ROS species [1,7,9,11,15,16,17]. More specifically, for the detoxification and oxidation of cytotoxic MG, another defense system (glyoxalase) which consists of glyoxalase I (Gly I) and glyoxalase II (Gly II) [9,18,19]. In addition to antioxidants, bioactive metabolites such as amino acids, proteins, carboxylic acids, and sugars are crucial for the control and alterations of a variety of key mechanisms in response to abiotic stresses [7,20,21,22,23,24]. In wheat seedlings, it has been observed that the presence of AsA and GSH enhances osmoregulation, cellular water and nutrient status, photosynthetic rate, and overall plant growth and productivity [25,26]. Previous research showed that the AsA–GSH cycle helps plants regulate a number of physiological and molecular pathways in addition to scavenging ROS [25,27,28,29].

Over the past few decades, various techniques have been introduced to minimize hazardous environmental effects including drought, salinity, heavy metals, heat, and cold on plants [30,31]. However, biochar (BC), which is an eco-friendly bio-stimulant that mostly increases crop production and alleviates the hazardous effects of different abiotic stresses, has been neglected until now. It also acts as both soil fertilizer and conditioner, which results in an improved crop yield and soil physio-chemical and biological properties, respectively [32]. Additionally, BC is essential for the growth and development of plants because it controls ion homeostasis, physiological processes, biochemical machinery, and metabolic profile, which minimizes oxidative damage [32,33,34,35,36]. However, the stress-relieving role of BC has not been thoroughly studied yet, especially under ROS–MG-induced oxidative stress in plants.

*Leptocohloa fusca,* also known as Kallar grass, is a C4 plant which has a significant role in livestock and dairy development without N-fertilization. This species has a particular importance since it acts as a soil-fixative species and it can be used as fodder in saline soils [37]. However, its productivity is also challenged by several abiotic stresses including drought stress. Therefore, the current research was designed to gain a better understanding of ROS–MG-induced oxidative damage in *L. fusca* and its mitigation by BC-amended soil. The objectives of the current study are: to investigate the physiological, biochemical, and metabolic responses of *L. fusca* under drought stress; to examine the role of bioactive metabolites in alleviating drought stress in *L. fusca*; to explore the ameliorative effects of BC under ROS– MG-induced oxidative damage.

## 2. Material and Methods

### 2.1. Experimental Setup

A 4-month-long pot experiment was conducted at a private agriculture farm near Lodhran–Khanewal Expressway, Punjab, Pakistan (29.79″ N, and 71.74″ E) using garden soil, and *L. fusca* (Kallar grass) was used as plant material. Organic amendment (biochar) was used as soil amendment. Before the start of the experiment (1st week of July 2021), two levels of biochar (BC-1 = 15 mg/kg soil and BC-2 = 30 mg/kg soil) were taken and thoroughly mixed with the garden soil. The mixture of BC + garden soil (6 kg of soil/pot) was filled in plastic bins. The pots were kept in normal semi-field condition under direct sunlight for 4–5 weeks. Soil physio-chemical properties are mentioned in Table 1. During the 6th week (2nd week of August 2021), *L. fusca* seeds were sown in plastic pots and properly irrigated until they reached the stage of 2–3 true leaves. During the 8th week of the experiment, plants were thinned to 3 plants per pot and drought stress (100%, 70%, and 30% FC) was applied to the plants. Pots containing growing medium and plants were inspected and irrigated daily in measured quantity of water to maintain respective FC throughout the experiment. The following were the treatments of the drought stress: FC-1 = 100% (control), FC-2 = 70%, FC-3 = 30%. A total of 9 treatments with different combinations were applied as follows: T0 = 100% (FC-1), T1 = 70% (FC-2), T2 = 30% (FC-3), T3 = FC-1 + BC-1, T4 = FC-1 + BC-2, T5 = FC-2 + BC-1, T6 = FC-2 + BC-2, T7 = FC-3 + BC-1, T8 = FC-3 + BC-2. During the 16th week, plants were harvested, and data was collected as per the protocols described below. The physiological and biochemical analyses were performed at the agriculture extension and research office (29.81″ N, and 71.73″ E), Pakistan.

### 2.2. Soil Preparation and Analysis

Soil samples were air-dried for 2 days, and impurities were removed by sieving the samples to ensure a constant particle size of 2 mm. Soil physico-chemical properties including pH, electrical conductivity (EC), total soluble salts (TSS), and soil salinity level were measured by using a pH meter (Kent Eil 7015, Amesbury, MA, USA), conductivity meter (Model 4070, U.S. Salinity Lab, Riverside, CA, USA) by using the digital Jenway method, and salinity sensor (for NaCl content) (Soil Moisture Equipment Corporation, Goleta, CA, USA), respectively (Table 1). Total nitrogen (TN) and soil total carbon (TC) were determined through the Elemental analyzer (Thermo Scientific™, Waltham, MA, USA).

### 2.3. Morphological Characteristics

All the morphological characteristics including shoot length (SL), root length (RL), shoot fresh weight (SFW), and root fresh weight (RFW) were measured by using a standard measuring scale and digital weighing balance. After recording the fresh weights, plants samples were placed in a dry oven at 70 °C to obtain a constant root dry weight (RDW) and shoot dry weight (SDW).

### 2.4. Nutrient Uptake

The contents of sodium (Na^+^), chloride (Cl^−^), nitrate (NO_3_^−^), potassium (K^+^), calcium (Ca^2+^), and magnesium ion (Mg^2+^) were measured from dry shoots and roots tissues by previously reported methods [38,39]. From ground homogenous dry plant samples, 0.1 g was digested with the acid mixture (HNO_3_: HClO_4_; 5:1). The Na^+^, Cl^−^, NO_3_^−^ K^+^, Ca^2+^, and Mg^2+^ contents were determined from the digested solution by using an atomic absorption spectrophotometer (AA-7000, Shimadzu, Japan).

### 2.5. Physiological Parameters

Total chlorophyll content (TC), stomatal conductance (gs), and photosynthetic rate (Pn) were analyzed by following standard procedures. The gas exchange parameters were recorded using a LiCor portable photosynthesis system (model LI-6200) calibrated at 800 mmol photons m^−2^ s^−1^ irradiance and 330–370 mmol CO_2_ mol^−1^. Stomatal conductance was measured on the adaxial leaf surface using a LiCor LI-1600 steady-state porometer [40]. Total chlorophyll (TC) was calculated using the previously developed equation [41].

### 2.6. ROS and MG Content

Hydrogen peroxide (H_2_O_2_) content was determined by a peroxidase dependent assay adopting the earlier method [42]. The presence of hydroxyl radicals (OH^−^) in the root and shoot samples was determined following the published procedure [43]. Plant samples were homogenized in 1.2 mL of 50 mM sodium phosphate buffer solution at pH 7.0, and centrifuged at 12,000 rpm for 10 min at 4 °C. Afterwards, 0.5 mL supernatant was mixed in 1 mL of 25 mM sodium phosphate buffer solution containing 2.5 mM 2-deoxyribose solution and incubated at 35 °C in the dark for 1 h. Then, the mixture was mixed with 1 mL glacial acetic acid and 1 mL of 1% thiobarbituric acid (TBA; Sigma, USA) and boiled for 10 min before immediately cooling in an ice bath. The absorbance was then measured at 532 nm.

Superoxide radicals (O_2_^−^) were detected by transferring the plant samples into 0.2% nitro blue tetrazolium chloride (NBT) dissolved in 50 mM sodium phosphate buffer at pH 7.5. An insoluble formazan compound of dark-blue color appeared when NBT reacted with O_2_^−^. The samples were then moved to a bleaching solution to remove the chlorophyll. Afterwards, samples were ground in 0.1% acetic acid solution and centrifuged at 10,000 rpm for 10 min, with the absorbance noted at 560 nm [44].

For Methylglyoxal (MG) content, fresh samples (100 mg) were ground in distilled water and centrifuged at 11,000 rpm for 10 min. After that, 100 μL of 5 M perchloric acid solution and 250 μL of 7.2 mM and 1,2-diaminobenzene solution were added to 650 μL of supernatant. The absorbance at 336 nm was recorded with a spectrophotometer (JASCO- V 530) [45].

For the determination of the lipid peroxidation, the amount of MDA formed by the TBA reaction was measured [46]. Ground samples of shoots and roots were centrifuged at 10,000 rpm for 5 min. To 1.0 mL of supernatant, in a separate test tube, 4.0 mL of 0.5% TBA was added. The mixture was heated at 95 °C for 30 min, then cooled in ice-cold water and later centrifuged at 5000 rpm for 5 min. Absorbance was measured at 532 nm and corrected for unspecific turbidity by subtracting the value at 600 nm.

### 2.7. Antioxidant Activity

Enzymatic and non-enzymatic antioxidants such as catalase (CAT), peroxidase (POD), superoxide dismutase (SOD), ascorbate peroxidase (APX), gluthione (GSH) and gluthione disulfide (GSSG), glutathione peroxidase (GPX), gluthione reductase (GR), gluthione S-transferase (GST), glyoxalase I (Gly I), glyoxalase II (Gly II), ascorbic acid (AsA), dehydroascorbate (DHA), monodehydroascorbate reductase (MDHAR), and dehydroascorbate reductase (DHAR) were measured, extracted, and assayed by the using the following protocols [39,47,48,49,50,51,52,53,54,55], respectively.

### 2.8. Metabolite Extraction

All extraction of metabolites such as amino acids, organic acids, and sugars in both roots and shoots of *L. fusca* grass were carried out by the earlier documented procedure [56]. Sample volumes of 1 μL were analyzed with a Trace GC gas chromatograph coupled to a PolarisQ ion trap mass spectrometer equipped with an AS2000 auto sampler (Thermo Electron, Dreieich, Germany). Derivatized metabolites were evaporated at 250 °C in splitless mode and separated on a 30 m × 0.25 mm RTX-5MS capillary column with a 0.25 mm coating equipped with an integrated 10 m guard column (Restak, Bad Homburg, Germany). A helium carrier gas flow was adjusted to 1 mL/m. The interface temperature was set to 250 °C and the ion source temperature to 220 °C. The oven temperature was kept constant for 3 min at 80 °C after each analysis. Mass spectra were recorded at 1 scan/s with a scanning range of 50 to 750 *m*/*z*. Metabolites were identified by comparison with pure standard (Sigma-Aldrich, Saint Louis, MO, USA). In addition, the freely available Golm Metabolome Database [57] was of particular help in identifying several metabolites. All identified compounds matched the references by mass spectral data and chromatographic retention time. Relative levels of selected metabolites were determined automatically by integrating the peak areas of selective ions [58] with the processing setup implemented in Xcalibur 1.4 software (Thermo Electron, Dreieich, Germany). Relative response ratio was calculated by normalizing the respective peak areas to the peak area of the internal standard ribitol and dividing the value by the dry weight of the sample. Measurements were performed in technical duplicates for each of the three replicates of control and the BC-amended and non-amended plants.

### 2.9. Statistical Analysis

A pot experiment was conducted with 9 treatments in total (3 replications each). All the data were analyzed statistically and significant differences were calculated by the one-way analysis of variance technique (ANOVA) under complete randomized design (CRD) using the SPSS software version 16.0 (Chicago, IL, United States). The least significance difference (LSD) test was applied to compare means at a 5% probability level. Duncan’s Multiple Range Test (DMRT) was used as a post hoc mean-separation test (*p* < 0.05) using SPSS statistics (16.0) software.

## 3. Results

### 3.1. Interactive Effect of Drought and Biochar on Morphological and Physiological Attributes of L. fusca

Both drought levels substantially reduced the morphological and physiological parameters of *Leptocohloa fusca* (Table 2). The morphological attributes such as SFW, RFW, SDW, and RDW considerably reduced by 66%, 42%, 68%, and 43% under 70% FC, and 79%, 76%, 90%, and 85% at 30% FC, respectively. Nevertheless, BC-2 application significantly enhanced SFW and SDW by 1.4- and 1.8-fold under 70% FC, and 2.2- and 6.8-fold under 30% FC, respectively. BC-1 treatment increased the RDW by 1.4-fold under 70% FC, whereas BC-2 amendment enhanced it by 2.9-fold at 30% FC as compared to unamended soil (Table 2). Furthermore, BC amendment had no discernible effect on RFW under any of the drought conditions.

TC, Pn, and gs were reduced by 39%, 35%, and 21% following 70% FC, and 63,% 67%, and 43% after 30% FC, respectively (Figure 1). Significant improvements were noticed only in gs after BC-2 application under 30% FC, while TC showed substantial increment after 70% FC by BC-2 application.

### 3.2. Alleviation of Oxidative-Stress Biomarkers by Biochar under Drought Stress

Under water-limited conditions, *L. fusca* plants substantially produced reactive oxygen and methylglyoxal species including H_2_O_2_, OH^−^, O_2_^−^, MDA, and MG. At 70% FC, H_2_O_2_, OH^−^, O_2_^−^, MDA, and MG increased by 1.9-, 2-, 1.7-, 2.3-, 3-fold, while after 30% FC, this elevation was 5.1-, 3-, 2.3-, 3.3-, 4.4-fold, respectively (Figure 2). However, BC-1 application significantly reduced the reactive species of *L. fusca* by 31%, 25%, 12%, 24%, and 30% after 70% FC, and 32%, 24%, 22%, 22%, and 25% following 30% FC, respectively (Figure 2). Additionally, the plants grown under BC-2 amended soil also experienced similar and/or more inhibition in oxidative-stress biomarkers.

Similarly to roots, the shoots of *L. fusca* plants experienced an abrupt amplification in the H_2_O_2_, OH^−^, O_2_^−^, MDA, and MG levels after drought stress, which was shown by the 2-, 2.6-, 2.1-, 2.2-, 2.8-fold elevation following 70% FC, and 5-, 5-, 3.9-, 3.3-, and 4-fold enhancement after 30% FC, respectively. In contrast, 66%, 26%, 23%, 42%, and 46% reduction after 70% and 21%, 24%, 17%, 34%, and 43% inhibition after 30% FC was noticed in the H_2_O_2_, OH^−^, O_2_^−^, MDA, and MG levels of the plants grown under BC-1 amended soil, respectively. Similarly, the addition of BC-2 led to the reduction in H_2_O_2_, OH^−^, O_2_^−^, MDA, and MG levels up to 60%, 54%, 66%, 63%, and 84% under 70%, and 36%, 47%, 26%, 37%, and 63% following 30% FC, respectively (Figure 2).

### 3.3. Enhancement in Nutrient Uptake by Biochar under Drought Stress

Based on hierarchical clustering, the separated group of control plants showed the influence of drought and biochar treatments on nutrient uptake in both roots and shoots of *L. fusca* plants. The roots and shoots of *Leptocohloa fusca* showed a significant reduction in Na^+^, Ca^2+^, Mg^2+^, NO_3_^−^, K^+^, and Cl^-^ under both the drought levels. In roots, Na^+^, Ca^2+^, Mg^2+^, NO_3_^−^, K^+^, and Cl^−^ faced reductions of 46%, 44%, 18%, 41%, 33%, and 61% following 70% FC, and 71%, 78%, 65%, 66%, 59%, and 55% inhibition under 30% FC, respectively (Figure 3). In contrast, BC-1 application showed an increment of 1.2-, 1.3-, 1.1-, 1.8-, 1.07-, and 1.3-fold when the plants were cultivated under 70% FC, while this improvement was 1.3-, 1.6-, 1.5-, 2-, 1.5-, and 1.5-fold for the plants under 30% FC. Similarly, BC-2 application exhibited 1.5-, 1.5-, 1.2-, 2.3-, 1.3-, and 2.8-fold increases following 70% FC and 1.8-, 2-, 1.8-, 3-, 1.9-, and 2.1-fold increases after 30% FC, respectively.

Similar to the roots, the shoots of *L. fusca* presented a considerable reduction in Na^+^, Ca^2+^, Mg^2+^, NO_3_^−^, K^+^, and Cl^−^ contents, which was 26%, 10%, 51%, 38%, 41%, and 51% following 70% FC and 60%, 43%, 84%, 58%, 72%, and 74% under 30% FC, respectively. The nutrient uptake in *L. fusca* shoots was enhanced by BC-1 at 70 % FC (1.3-, 1.05-, 1.2-, 1.2-, 1.4-, 1.2-fold), and at 30% FC (1.4-, 1.2-, 2.5-, 1.4-, 2.1-, 1.8-fold), respectively. Similar improvements were exhibited by BC-2 at all FC levels.

### 3.4. Biochar-Dependent Changes in Antioxidant Activities under Drought Stress

According to hierarchical clustering, the control plants made separate group than the treated plants, which showed the obvious effect of treatments in both roots and shoots of *L. fusca* plants (Figure 4). The studied antioxidants showed different trends since Gly I, Gly II, SOD, CAT, POD, GPX, GSH, AsA, GR, DHAR, and DHA reduced in roots by 44, 16, 24, 5, 39, 14, 38, 25, 46, 17, and 35% following 70% FC, and 88, 73, 58, 25, 66, 24, 60, 45, 69, 30, and 63% after 30% FC, respectively. However, GST and MDHAR showed a slight increment of 2- and 1.3-fold at 30% FC and 3.1- and 1.6-fold at 30% FC, respectively. Contrary to others, APX and GSSG in roots showed a different pattern of growth under both stress conditions. APX and GSSG increased under 70% FC, whereas both of them were down-regulated at 30% FC. However, BC amendment enhanced the antioxidant activity in roots with few exceptions (Gly II under 70% FC and GST under 30% FC in the case of BC-1). The plants treated with BC-1 presented 1.5-, 2.4-, 2.5-, 1.1-, 1.2-, 1.4-, 2-, 1.7-, 1.1-, 1.5-, 2.5-, 1.2-, 2-, and 2.1-fold up-regulation in the activities when grown under 70% FC, and 3.5-, 3-, 5.6-, 2-, 2-, 1.1-, 1.2-, 2-, 2-, 1.2, 1.6-, 1.07-, 2-, and 2.4-fold stimulation for Gly I, SOD, CAT, POD, APX, GPX, GSH, GSSG, AsA, GR, DHAR, MDHAR, and DHA activities, respectively. BC-2 also exhibited similar stimulation in the activities of antioxidants for both drought levels.

Similar to roots, antioxidant (Gly I, Gly II, SOD, POD, APX, GPX, GSH, GSSG, AsA, GR, DHAR, and DHA) activities experienced a decline in shoots after 70% FC (6, 42, 35, 36, 8, 31, 30, 6, 22, 44, 29, and 50%) and under 30% FC (8, 75, 75, 80, 48, 60, 70, 35, 45, 80, 55, and 83%), respectively. CAT, MDHAR, and GST exhibited different and interesting trends. Moreover, BC improved antioxidant activity significantly in shoots of *L. fusca*. BC-1 application improved Gly I, Gly II, SOD, CAT, POD, APX, GPX, GSH, GSSG, AsA, GR, GST, DHAR, MDHAR, and DHA activities up to 1.8-, 3.3-, 2.6-, 1.9-, 1.3-, 1.5-, 2.2-, 1.8-, 1.2-, 1.06-, 1.7-, 2.4-, 1.3-, 2.2-, and 1.4-fold following 70% FC, and 3-, 8-, 2-, 3-, 1.4-, 1.2-, 1.3-, 1.6-, 1.4-, 1.2-, 2.8-, 2.6-, 1.4-, 2.3-, and 4.5-fold under 30% FC, respectively. Similarly, BC-2 substantially enhanced Gly I, Gly II, SOD, CAT, POD, APX, GPX, GSH, GSSG, AsA, GR, GST, DHAR, MDHAR, and DHA activities.

### 3.5. Modifications in Metabolite Levels by Biochar under Drought Stress

Based on hierarchical clustering, the separated group of control plants showed the influence of drought and biochar treatments on metabolites in both roots and shoots of *L. fusca* plants (Figure 5). Amino acids such as Ala, Gly, Leu, Pro, Val, Trp, Tyr, Lys, Ser, Thr, Cys, Met, Asn, Gln, His, Arg, Glu, and Asp reduced up to 44, 50, 31, 26, 21, 51, 52, 60, 30, 36, 40, 58, 55, 51, 34, 57, 65, and 43% under 70% FC, and 74, 97, 61, 47, 59, 77, 76, 81, 78, 65, 54, 82, 93, 89, 57, 88, 89, and 79% in roots following 30% FC, respectively. In contrast, BC amendment considerably improved the amino acid levels in *L. fusca* roots, as evidenced by 1.4-, 3-, 1.6-, 1.2-, 1.7-, 3.4-, 1.7-, 1.7-, 2-, 1.3-, 2.2-, 1.4-, 3-, 1.2-, 2.2-, 2.9-, and 1.6-fold increment in Ala, Gly, Leu, Val, Trp, Tyr, Lys, Ser, Thr, Cys, Met, Asn, Gln, His, Arg, Glu, and Asp contents when plants were supplied with BC-1 under 70% FC, and 3-, 4.3-, 2.1-, 1.5-, 2.5-, 2.7-, 3.3-, 1.7-, 2.4-, 1.8-, 1.5-, 4-, 6-, 3.3-, 1.6-, 3.1-, 4.4-, and 2.8-fold up-regulation in Ala, Gly, Leu, Pro, Val, Trp, Tyr, Lys, Ser, Thr, Cys, Met, Asn, Gln, His, Arg, Glu, and Asp after 30% FC, respectively. In addition, similar up-regulation of the examined amino acids was found by the addition of BC-2 (Figure 5).

Organic acids including CA, FA, LA, SA, BA, MA, PA, OA, GA I, and GA II decreased by 30, 56, 31, 32, 41, 29, 42, 63, 27, and 32% in roots following 70% FC, while69, 75, 71, 72, 63, 72, 74, 98, and 56% after 30% FC, respectively (Figure 5). However, BC-1 amendment considerably improved these organic acids by 1.2-, 2.9-, 1.6-, 1.6-, 1.8-, 1.4-, 1.8-, 2-, 1.2-, and 1.2-fold when the plants were cultivated under 70% FC, while this enhancement was 1.7-, 4.4-, 1.8-, 2.6-, 2.3-, 1.4-, 2.8-, 24.8-, 1.4-, and 3.1-fold in the plants treated with 30% FC, respectively. A similar improvement in the investigated organic acids was noticed when the plants were grown in BC-2-amended soil under both drought-stress levels (Figure 5).

Moreover, sugars such as ribose, fructose, maltose, xylose, rhamnose, melibiose, and pinitol reduced by 41, 47, 52, 42, 52, 48, and 42% in roots of *L. fusca* plants under 70% FC, and 71, 70, 89, 72, 88, 63, and 68% inhibition when the plants were treated with 30% FC, respectively (Figure 5). However, the BC amendment (BC-1) remarkably elevated sugar (fructose, maltose, xylose, rhamnose, melibiose, and pinitol) levels by 1.2-, 1.3-, 1.3-, 2-, 1.4-, and 1.3-fold in the plants treated with 70% FC, and ribose, fructose, maltose, xylose, rhamnose, melibiose, and pinitol showed 1.1-, 1.4-, 2.3-, 1.7-, 5.2-, 1.3-, and 1.5-fold enhancement in the plants grown in 30% FC, respectively. In addition, BC-2 also led to substantial stimulation of the sugar contents in *L. fusca* plants grown under both drought-stress levels (Figure 5).

Additionally in shoots, the amino acid content including Ala, Gly, Leu, Pro, Val, Trp, Tyr, Lys, Ser, Thr, Cys, Met, Asn, Gln, His, Arg, Glu, and Asp declined by 37, 42, 31, 17, 31, 43, 41, 40, 18, 30, 24, 27, 27, 35, 28, 53, 39, and 43% by 70% FC, and this reduction was 64, 77, 71, 76, 81, 84, 72, 73, 79, 62, 53, 78, 76, 86, 53, 77, 82, and 70% after the employment of 30% FC, respectively (Figure 6). Similar to the up-regulation of amino acids in roots, the shoots of *L. fusca* plants presented stimulation of amino acid levels by both levels of BC under both drought conditions.

The shoots of *L. fusca* plants experienced 17, 25, 17, 39, 29, 26, 32, 55, 32, and 19% inhibition in organic acid content after 70% FC, and a 58, 75, 65, 68, 58, 71, 63, 85, 67, and 67% decline in CA, FA, LA, SA, BA, MA, PA, OA, GA I, and GA II following 30% FC, respectively. However, the BC amendment resulted in the up-regulation of all the examined organic acids under both drought-stress levels (Figure 6).

Furthermore, the sugar content in shoots of *L. fusca* plants exhibited the similar response against drought stress and biochar amendment. The sugars ribose, fructose, maltose, xylose, rhamnose, melibiose, and pinitol reduced by 42, 34, 49, 38, 58, 55, and 44% at 70% FC, and 45, 74, 86, 74, 78, 70, and 74% at 30% FC compared to unamended soil, respectively (Figure 6).

## 4. Discussion

Drought stress is one of the hazardous environmental factors influencing plant growth and development by modifying physiological and biochemical systems [59,60]. However, the use of fodder or forage grasses in such neglected soils is the ideal approach to make use of barren or water-stressed soils. *Leptocohloa fusca* has been considered an ideal plant for saline soils but it was not previously studied under drought-stress conditions. Hence, the current research was carried out to explore the potential of *L. fusca* under water stress and the effects of BC amendments on morpho-physiological and biochemical characteristics. The results of the current study revealed that, under drought stress, morpho-physiological attributes substantially reduced with a progression in drought severity. These findings were parallel with previous studies [21,61,62,63,64,65]. In addition, organic-soil amendment (BC) significantly improved plant growth including SL, RL, RFW, SFW, RDW, and SDW. These results were found indirectly in accordance with the previous findings, wherein organic soil amendments considerably reduced the toxic effects of abiotic stress, including heavy metals, drought, and salinity in different plant species [21,32,66,67,68,69]. All the physiological parameters such as TC, Pn, and gs reduced in the current experiment. Some previous studies were found in line with the current results [65,70,71]. BC amendment considerably reduced drought stress and improved the physiological attributes of *L. fusca*. Previous studies on *Triticum aestivum* L. [72], *Beta vulgaris* [35], and *Zea mays* [73] were in accordance with our current results.

Mineral nutrients are the building blocks of essential organic molecules, for example, amino acids, enzymes, proteins, organic acids, DNA, and RNA [2]. Water scarcity impairs the nutrient uptake in plants and hence affects many biological functions which ultimately influence plant overall growth and development [59]. Some research also showed that under extreme or persistent drought-stress conditions, nutrient supplementation could not improve the plant’s nutritional state [74]. Soil fertility and plant nutrient uptake has been a topic of discussion among researchers. However, recent studies suggested that nutrient supply under drought stress does not have any significant impacts on plant nutritional status, but it mainly depends on water availability [75]. The current results showed a visible difference in nutrient uptake such as Na^+^, Cl^−^, Ca^2+^, Mg^2+^, K^+^, and NO_3_^−^ in *L. fusca* roots and shoots following drought stress. Some previous studies were consistent with the current results [21,59,63,64,65,71,76,77]. In contrast, BC application significantly improved Na^+^, Cl^−^, Ca^2+^, Mg^2+^, K^+^, and NO_3_^−^ uptake in *L. fusca*’sroots and shoots. Previous studies confirmed the current findings, wherein BC helped in plant nutritional uptake by improving soil physio-chemical properties and soil water-holding capacity [33,36,78,79]. With limitations to gene involvement in response to water stress, our results suggested that the significant decline in physiological, biochemical, and metabolic response in *L. fusca* might be due to the decrease in plant nutrient uptake. However, stress-responsive genes and their mechanisms in plant physiological, biochemical, and metabolic response should be investigated to better understand responses of *L. fusca* against drought stress.

The accumulation of ROS and their damaging effects on cellular organelles are the core elements of plants exposed to abiotic stressors [80]. ROS such as MDA, H_2_O_2_, O_2_^−^, OH^−^, ^1^O_2_, etc., are the primary products in plant metabolism. However, in higher amounts these reactive species cause serious damage to plant cells which ultimately leads to cell death. In the current experiment, the accumulation of ROS caused oxidative damage to *L. fusca* plants under water-stress conditions and showed toxicity symptoms such as leaf wilting, necrosis, and denaturation of light-dependent photosynthetic systems. A plethora of studies have confirmed our current findings where MDA, H_2_O_2_, O_2_^−^, and OH^−^ increased with an increase in drought severity [7,8,9,17,19,28,70]. In contrast, BC application substantially reduced ROS-mediated stress in *L. fusca,* which was also confirmed by the earlier studies [15,32,35,68,81,82,83]. As well as ROS, recent studies revealed that, as a result of excessive lipid peroxidation, some reactive carbonyl species (RCS) are also produced, which cause oxidative injury to plant cells [11,84]. Excessive MG production in roots and shoots of *L. fusca* confirmed that RCS was produced downstream of ROS and caused irreversible oxidative damage to plant cells in response to drought stress, and similar outcomes were found in previous experiments [11,84,85,86]. To combat ROS–RCS-induced oxidative damage, plants are well-equipped with protective mechanisms including antioxidants. In the current study, antioxidants including SOD, CAT, POD, APX, GSH, AsA, GST, MDHAR, DHAR, DHA, GPX, Gly I, and Gly II played a crucial role in alleviating ROS–MG-induced oxidative damage in *L. fusca* plants. Along with other antioxidant enzymes, GSH–AsA-cycle-related enzymes (Figure 6) substantially reduced H_2_O_2_, O_2_^−^, OH^−^, and MDA, levels. Gly I, and Gly II are two-step MG-detoxification pathways that convert MG to D-lactate using glutathione as a co-factor. Our results are broadly in line with previous findings [11,18,19,28,29,85,86], whereas, BC application considerably improved the antioxidants in *L. fusca* roots and shoots. These results are indirectly in line with previous studies wherein BC application alleviated oxidative stress in *Brassica chinensis, Carthamus tinctorius*, *Zea mays*, *Setaria italica*, and *Eisenia fetida* under different abiotic-stress conditions [34,66,87,88,89]. However, the current study provides new insights into ROS–MG-induced oxidative stress and its alleviation by BC application. However, MG-induced oxidative stress and its alleviation by organic soil amendments, especially BC application, is still at rudimentary stages and needs more scientific knowledge in future.

Another promising aspect of this research was its metabolomics profile, which revealed a drop in the majority of metabolites under water stress. Some previous studies showed that the metabolites have signaling effects toward stress tolerance in plants. However, this entirely depends on the plant species, type, and intensity of stress and soil fertility [2,22,23,84]. ROS oxidize amino acids and the alteration of this key organic component result in loss of given protein-mediated functions including metabolic, structural, transport, and regulatory activities, which ultimately leads to cell death in plants [2,90,91]. Three essential amino acids—tryptophan, tyrosine, and phenylalanine—produced by the shikimate pathway are essential for plant development, stress tolerance, and pest resistance [92]. Studies have demonstrated that, under water stress, maize and chickpea plants hyper-accumulated tryptophan, phenylalanine, proline, and tyrosine [83,93]. Contrary to these findings, the current study revealed that amino acid substantially reduced under drought stress in *L. fusca* plants [2,87,94,95,96,97,98]. Sugars are the most abundant group of organic molecules in plants [2,99]. Soluble sugars are the integral part of plant cells and take part in many vital activities such as antioxidant defense systems, and mechanically shape and support plant cells [2,90]. Furthermore, sugars such as sucrose, maltose, fructose, glucose, sorbitol, and deoxyribose may act as ROS-scavenging molecules. Their ROS-scavenging order is as follows: maltose > sucrose > fructose > glucose > deoxyribose > sorbitol [100,101]. However, the direct link between oxidative stress-induced modification to sugar and plant physiology under abiotic stress is still missing. In future, sugar-stress-related genetic study is also needed to better understand the scavenging or signaling role of sugars in plants under water-stress conditions.

In many plant species, organic acids are crucial for their ability to withstand drought stress [2,83]. Some earlier studies elaborated that lowered levels of organic acids result in long-term drought resistance in plants [102,103]. On the other hand, in the current experiment, a significant reduction in organic acids such as CA, FA, LA, SA, BA, MA, PA, OA, GA I, and GA II in both roots and shoots of *L. fusca* was noticed after drought stress [24,71,89,95,104,105,106,107]. This might be due to the high sensitivity of *L. fusca* plants to water-stress conditions. However, in future, a stress-responsive genetic study associated with the exceptional behavior of *L. fusca* is needed to have better understanding. With limitations to the stress-responsive genetic study of *L. fusca* plants under drought stress, the current findings confirmed that the *L. fusca* plant is a drought-sensitive plant species, as evidenced by the remarkable reduction in metabolites, which were then improved by BC amendment. These findings revealed a strong correlation between BC amendment and drought-stress tolerance in plants. These results confirmed the findings of some previous studies [32,66,67,68,88,108,109,110].

## 5. Conclusions

The findings revealed that water stress had a negative impact on the nutrient intake of *L. fusca*, which in turn hampered overall growth and development of *L. fusca*. Reduced nutrient uptake in *L. fusca* plants during drought stress interfered with several physiological and biochemical systems, altering overall plant growth and development. However, biochemical studies demonstrated that the AsA–GSH cycle, along with other antioxidants, had a substantial impact on physiological modifications and ROS-detoxification in *L. fusca* in response to drought stress (Figure 7). Another finding of this study is the presence of MG, a highly reactive dicarbonyl molecule that is formed downstream to ROS and is responsible for oxidative damage in plants under water stress. The glyoxalase system, which is involved in MG detoxification, is comprised of two enzymes, Gly I and Gly II, which completed the conversion of MG to D-Lactate/lactic acid by utilizing GSH as a co-factor (Figure 7). Furthermore, it was found that BC application substantially reduced ROS–MG-induced oxidative damage by promoting antioxidant activity, protein, and amino and organic acids, and reducing ROS and MG content, which in turn improved morphological and physiological growth in *L. fusca* under drought stress. However, the concept of ROS–MG-induced oxidative stress in plant molecular studies is still at a rudimentary stage, and further studies are needed at genetic levels which will unfold the roles of proteins, antioxidants, and genes involved in MG-induced oxidative stress and its detoxification mechanism in *L. fusca* under drought stress.

## Figures and Tables

**Figure 1 metabolites-13-00511-f001:**
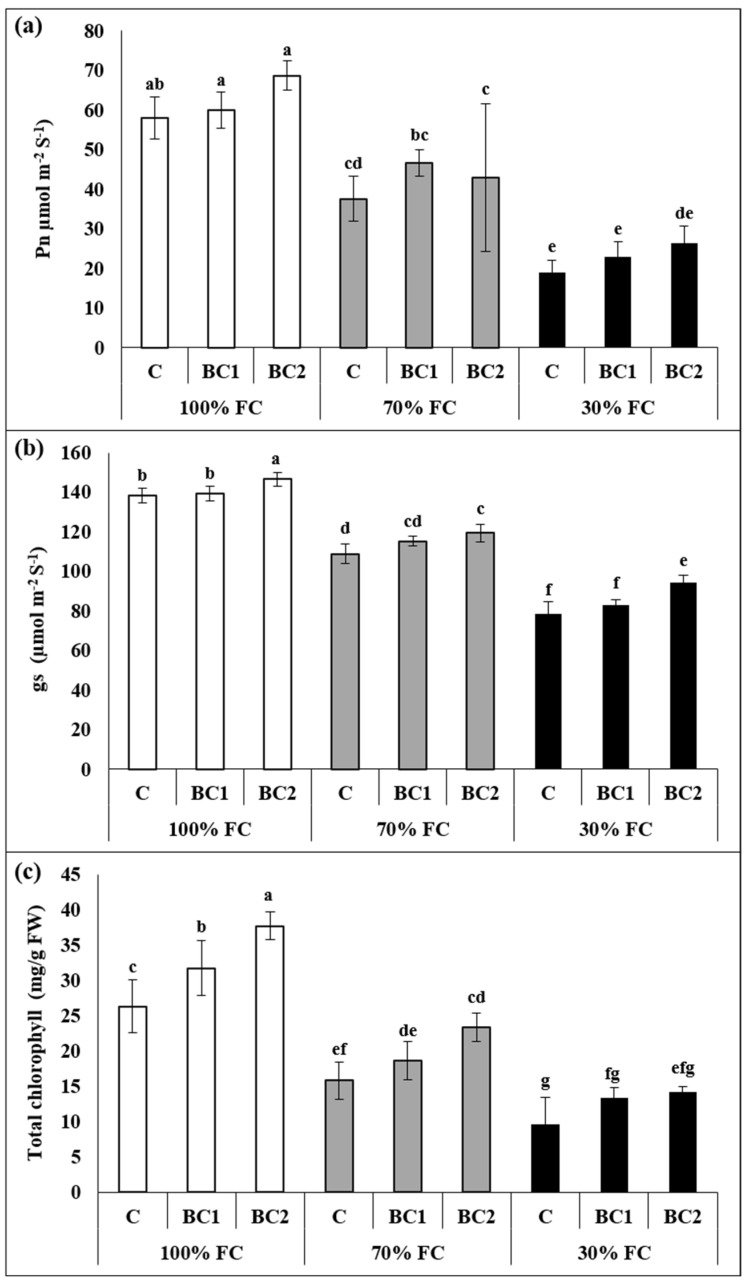
Improvement in the photosynthetic efficiency of *L. fusca* plants by biochar amendment under drought stress. (**a**): photosynthetic rate (Pn), (**b**): stomatal conductance (gs), (**c**): total chlorophyll. In addition, FC and BC represent field capacity and biochar, respectively. The values marked with different letters are significantly different from each other at *p* ≤ 0.05 levels in each parameter (ANOVA) followed by least significant difference (LSD) test.

**Figure 2 metabolites-13-00511-f002:**
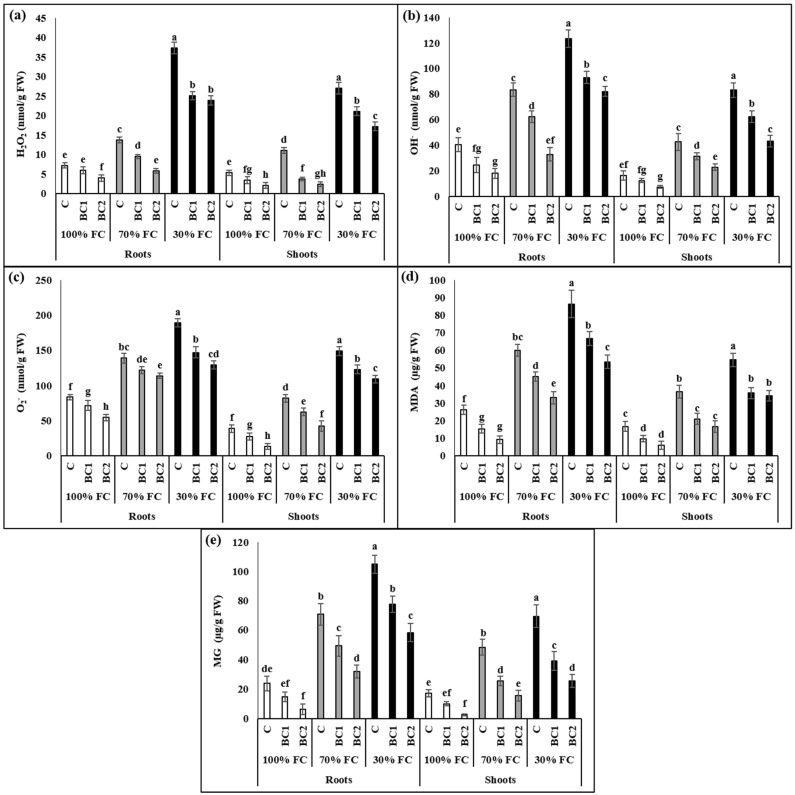
Reduction in the reactive species levels by biochar amendment in drought-stressed *L. fusca* plants. (**a**): hydrogen peroxide (H_2_O_2_), (**b**): hydroxyl ion (OH^−^), (**c**): superoxide ion (O_2_^−^), (**d**): malonaldehyde (MDA), (**e**): methylglyoxal (MG). In addition, FC and BC represent field capacity and biochar, respectively. The values marked with different letters are significantly different from each other at *p* ≤ 0.05 levels in each parameter (ANOVA) followed by least significant difference (LSD) test.

**Figure 3 metabolites-13-00511-f003:**
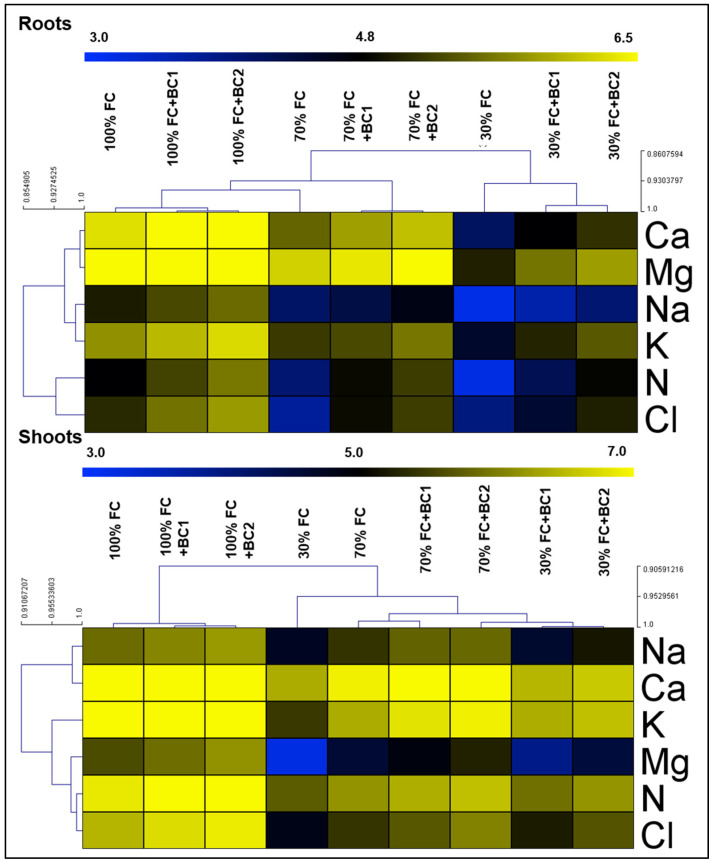
Biochar-induced improvement in the nutrient uptake of drought-stressed *L. fusca* plants. FC, BC, Na^+^, Ca^2+^, K^+^, Mg^2+^, NO_3_^−^, and Cl^−^ refer to field capacity, biochar, sodium, calcium, potassium, magnesium, nitrate, and chloride, respectively. The heatmaps were developed by Multiviewer Experiment software by using log-2 values.

**Figure 4 metabolites-13-00511-f004:**
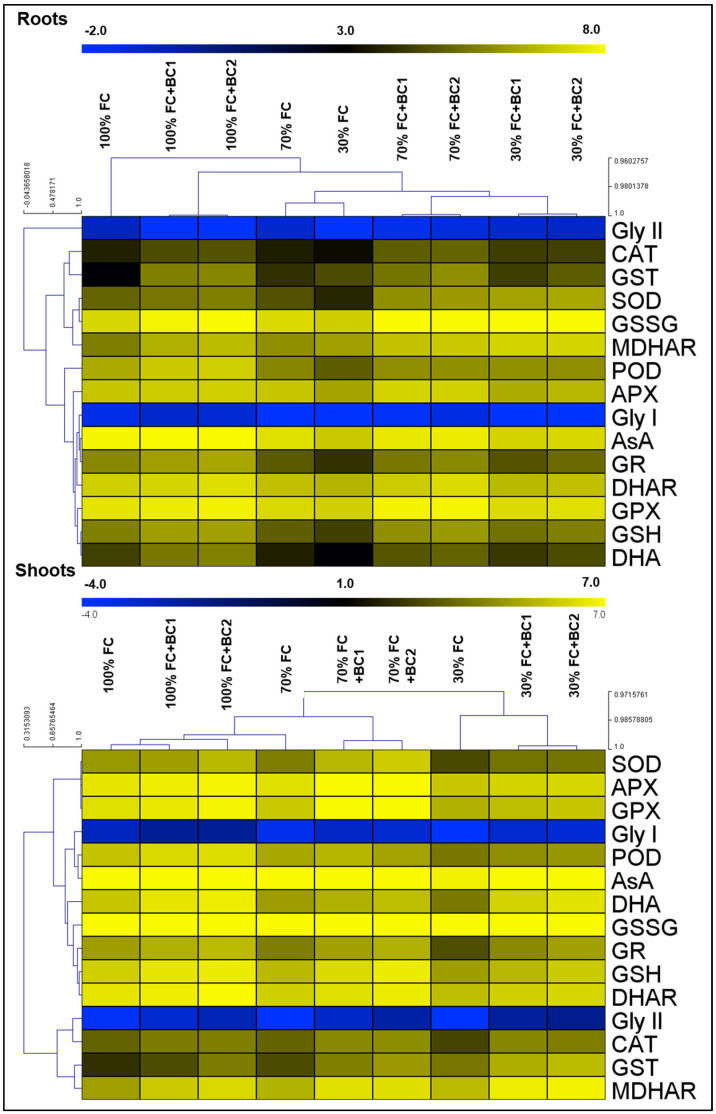
Biochar-dependent modifications in the antioxidant activities of drought-stressed *L. fusca* plants. Field capacity (FC), biochar (BC), superoxide dismutase (SOD), ascorbate peroxidase (APX), glutathione peroxidase (GPX), glyoxalase I (Gly I), peroxidase (POD), ascorbic acid (AsA), dehydroascorbate (DHA), glutathione disulfide (GSSG), glutathione reductase (GR), glutathione (GSH), dehydroascorbate reductase (DHAR), glyoxalase II (Gly II), catalase (CAT), glutathione-S-transferase (GST), and monodehydroascorbate reductase (MDHAR), respectively. The heatmaps were developed by Multiviewer Experiment software by using log-2 values.

**Figure 5 metabolites-13-00511-f005:**
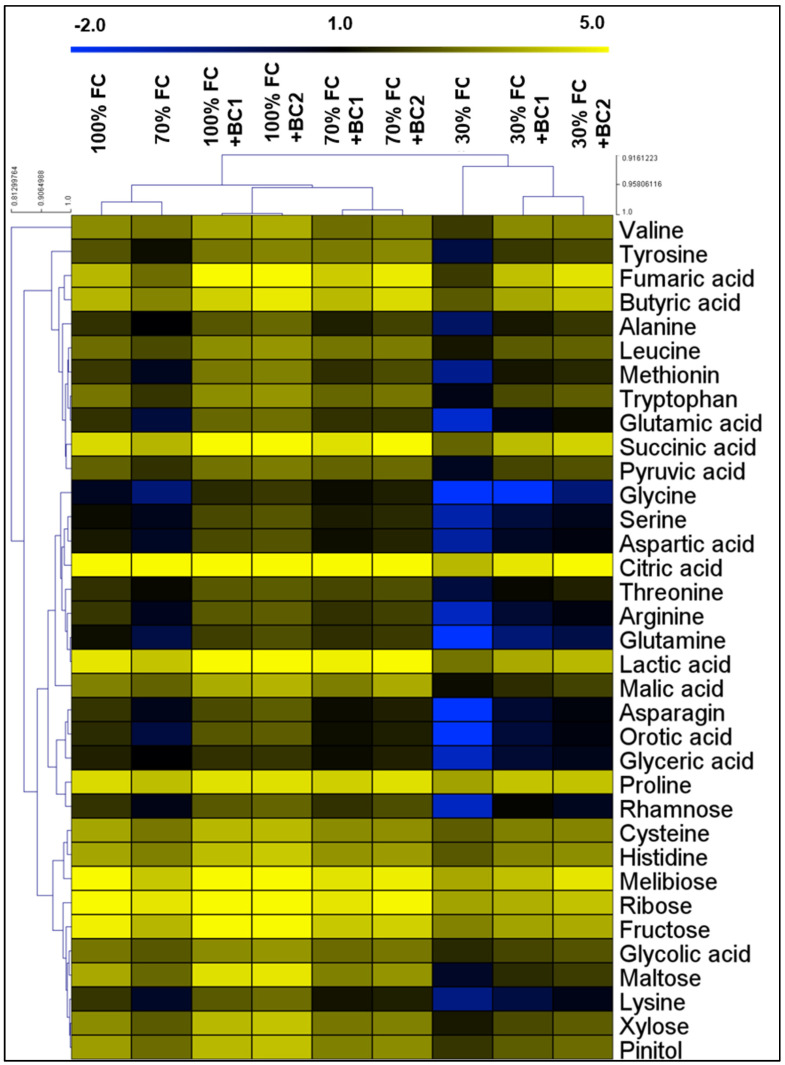
Biochar-dependent alterations in the metabolite levels of drought-stressed *L. fusca* roots. The heatmaps were developed by Multiviewer Experiment software by using log-2 values. In addition, FC and BC represent field capacity and biochar, respectively.

**Figure 6 metabolites-13-00511-f006:**
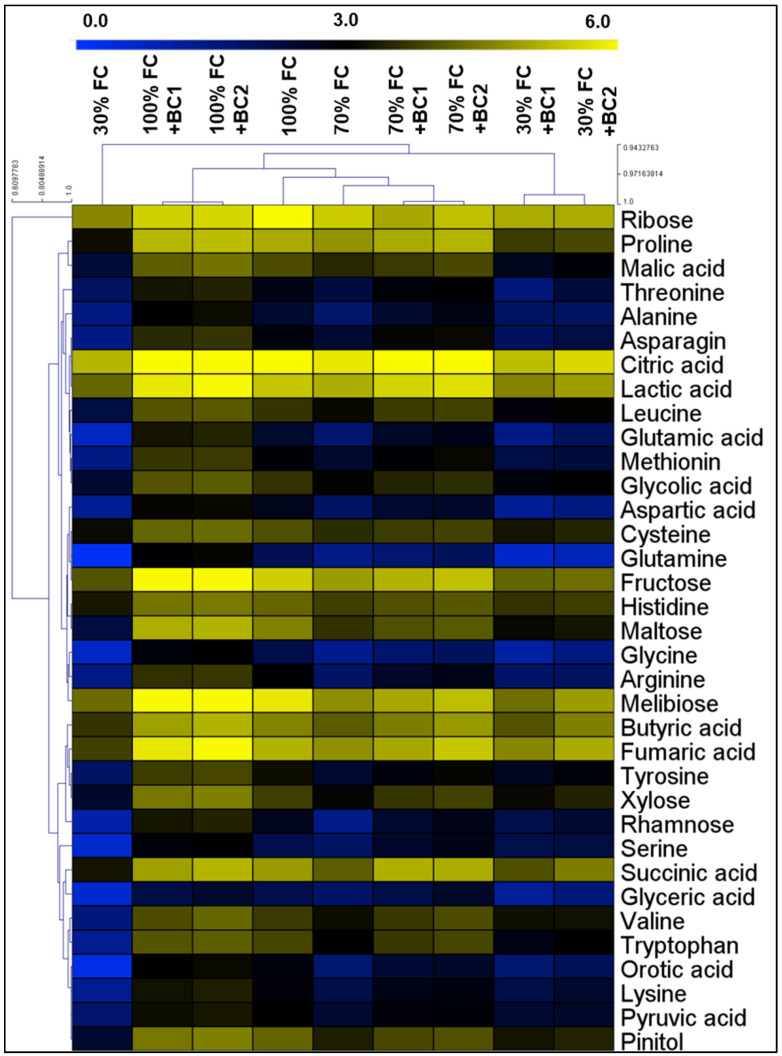
Biochar-dependent alterations in the metabolite levels of drought-stressed *L. fusca* shoots. The heatmaps were developed by Multiviewer Experiment software by using log-2 values. In addition, FC and BC represent field capacity and biochar, respectively.

**Figure 7 metabolites-13-00511-f007:**
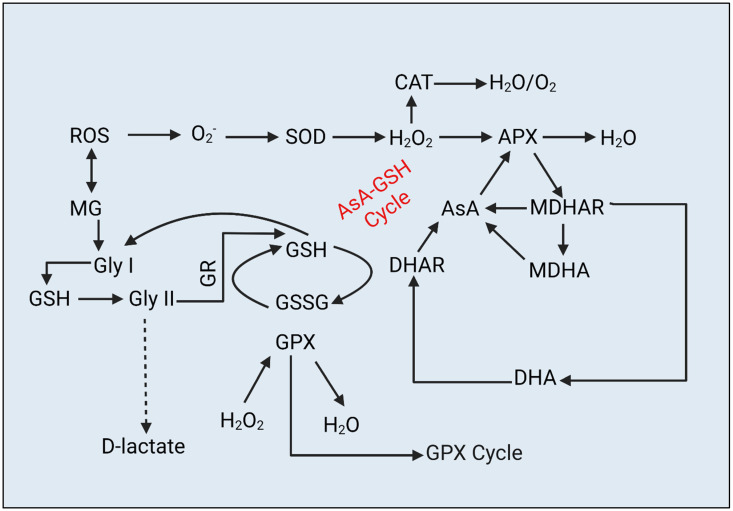
Schematic representation of the involvement of the ascorbate–glutathione (AsA–GSH) and Glyoxalase cycles in response to excessive reactive species (ROS and RCS) production under drought stress in *L. fusca* plants. The AsA–GSH cycle is typically involved in ROS scavenging, while the Glyoxalase cycle deals with MG detoxification in plants under stress. This figure was prepared by using Biorender software.

**Table 1 metabolites-13-00511-t001:** The physio-chemical properties of garden soil and biochar.

Parameters	Garden Soil	Biochar
pH	5.78	8.24
EC (mS/cm)	87.79	5.31
Organic matter (mg/kg)	54.93	135.47
Total soluble solids (TSS) (mg/kg)	49.65	178.95
Total carbon (mg/kg)	8.68	287.46
Total nitrogen (mg/kg)	7.94	9.56
NaCl content (mg/kg)	32.06	65.78

**Table 2 metabolites-13-00511-t002:** The hazardous effects of drought on morphological attributes of *Leptocohloa fusca* and its mitigation by biochar amendment.

Treatments	SFW (g)	RFW (g)	SDW (g)	RDW (g)
100% FC	43.2 4b	12.16 a	19.42 b	5.16 c
100% FC + BC1	46.51 b	11.89 a	21.61 b	6.65 b
100% FC + BC2	52.21 a	12.49 a	26.64 a	7.77 a
70% FC	14.28 d	6.97 b	6.18 e	2.93 ef
70% FC + BC1	17.88 cd	8.13 b	8.97 cde	4.21 cd
70% FC + BC2	20.77 c	8.84 b	11.16 cd	3.9 de
30% FC	8.84 e	2.83 c	1.81 f	0.74 h
30% FC + BC1	16.28 cd	3.81 c	8.27 de	1.78 g
30% FC + BC2	19.5 c	2.9 c	12.43 c	2.21 fg

The SFW (Shoot Fresh Weight), RFW (Root Fresh Weight), SDW (Shoot Dry Weight), and RDW (Root Dry Weight). In addition, FC and BC represent field capacity and biochar, respectively. The values marked with different letters are significantly different from each other at *p* ≤ 0.05 levels in each parameter (ANOVA) followed by least significant difference (LSD) test.

## Data Availability

Not applicable.

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
