# Peer review of "Biochar-Mediated Control of Metabolites and Other Physiological Responses in Water-Stressed *Leptocohloa fusca"

_metabolites, 2023, doi:10.3390/metabo13040511_

Round 1

Reviewer 1 Report

Dear Authors,

Although the concept of the paper "Biochar-mediated control of metabolites and other physiological responses in water-stressed Leptocohloa fusca" is quite impressive, it needs be significantly modified before publishing. I left some comments in the attached pdf file. 

Additional remark is to provide detailed information on statistical analyses performed and sample size utilised.

Author Response

Response to Reviewer 1

First of all, the authors wish to Thank for your valuable comments in improving the manuscript.

Abstract:

  1. Decipher the abbreviation

Response: The suggested changes have been made (line 33 and 40).

Introduction:

  1. References are required

Response: The relevant information references have been added in the revised MS (line number 58, 65, and 68). 

  1. Please rewrite the sentence specifying that this observation made on wheat seedlings.

Response:  Please see the line number 79-80 in Material and Method section. The suggested changes has been made in the revised MS (line 82-83).

  1. This statement is controversial, as providing essential nutrients depends on type and physio-chemical properties of biochar. Therefore, the sentence should be rewritten

Response: The sentence has been revised/modified (line 92).

Material and Method:

  1. Please double-check the units. The unit "mg/kg" was written in the abstract. Additionally, unify the representation of all units throughout the text, since form .../... is used at some points while ... ...-1 is used at others.

Response: The suggested changes have been made (line 112).

  1. Were the plant seeds sown in prepared pots? If no, provide the period when they were transferred, and exact date and place (GPS, etc), when and where the experiment was performed.

Response: Seeds were sown in prepared pots. The necessary information such as the GPS location of the experimental site and time duration of the experiment has been added in material and method section.

  1. 6 kg of pure soil or biochar-mixed soil? Please elaborate on this moment.

Response: Initially biochar (15 and 30 mg/kg) was mixed with garden soil and pots were filled with the biochar+garden soil mixture. Afterward, the seeds were sown in the pots. The sentence has been revised/ rewritten and more information has been added to the revised MS.

  1. Unify unit representation

Response: The units have been revised throughout the manuscript.

  1. Seems to be a quite low for garden soil.

Response: Nitrogen content in garden soil has been corrected in Table 1, page 3 in the revised MS.

  1. subscript. Correct this point throughout the text

Response: the suggested changes have been made throughout the MS.

  1. Provide manufacturer and country

Response: The necessary information regarding “spectrometer’s” manufacturer and country has been added in material and method section of the revised MS.

  1. lowercase. Correct this moment for all antioxidants mentioned below.

Response: the suggested changes have been made in the revised MS.

  1. Provide a full title

Response: full abbreviation of GSSG has been added in the revised MS.

Results:

  1. The statement is not correct as BC-1 did not improve all the parameters only RDW (Table 2), even BC-2 did not improve all the parameters. Rewrite the sentence. And check throughout the text statements you made, if significant difference was not found, the increase or decrease can not be stated.

Response: Dear reviewer, thank you for this valuable suggestion. We have tried our best to revise the MS as per your suggestion.

  1. Please provide a statement (in table note) about what analysis was performed, how many and what factors were considered as well as p-values for each parameter. As I understand Two-way ANOVA was performed with following factors: drought and biochar concentration

Response: There were two factors, drought and Biochar and we compared them with other which can be seen in the figure 1 and 2. One-way ANOVA was applied to compare the treatments in the roots and shoots separately.    

These are the letters of significance among the applied treatments for roots and shoots separately i.e. the treatments which were applied were compared in the roots separately and shoots separately. In addition one-way ANOVA was applied to compare the treatments.   

  1. Improvement was detected only for TC (Fig. 1c). Rewrite the statement

Response: The suggested corrections have been made in the revised MS.

  1. Please redraw the graph grouping the bars by drought stress, it would facilitate visual perception. Unify the units. Keep the same ordering of parameters as described in text. Double-check the Pn value for 70% FC + BC-2, SD is too wide. Provide p-values for all the analysis made for graph creation.

Response: Significant differences were calculated by one-way analysis of variance (ANOVA) using the SPSS statistics (16.0) software. The least significance difference (LSD) test was applied to compare means at a 5% probability level. Duncan’s Multiple Range Test (DMRT) was used as a post hoc mean-separation test (p < 0.05) using SPSS statistics (16.0) software.

  1. Please double-check the values. How did you calculate the decrease? Taking unamended treatments as controls?

Response: It depends upon with which treatment we are comparing for a specific statement. However, usually this comparison was made with the un-amended treatments as a control.

  1. There appears to be no significant increase, hence "substantially" cannot be said. Check the significance once again and revise the sentence.

Response: We have carefully checked this remark throughout the MS and corrected it where we found error.

  1. Heatmaps (Figs. 3, 4, and 5) appear to have been made without standardisation? Indicate the kind of standardisation used if data was standardised (row or column).

Response: All the values were converted into Log-2 and then prepared the heatmaps from that Log-2 values with the help of MultiExperiment Viewer software.

By following this scheme of heatmaps, we published multiple papers in the well reputed journals (Asghar et al. 2022b, a, 2023).   

  1. Same remark as above

Response: This remark has been corrected throughout the MS.

Conclusion:

1.Should be rewritten according to changes that would be made. And enrich the future conclusion with the main statements from results.

Response: The conclusion has been revised/improved including key findings of the current research and future directions in the revised MS.

In addition to all the suggested changes, some minor changes on line 93, 99, 100, 104, 111, 114, 125, 127, 176, 179, have been made in the revised MS.

References

Asghar MA, Balogh E, Ahres M, et al (2022a) Ascorbate and Hydrogen Peroxide Modify Metabolite Profile of Wheat Differently. J Plant Growth Regul. https://doi.org/10.1007/s00344-022-10793-0

Asghar MA, Balogh E, Szalai G, et al (2022b) Differences in the light‐dependent changes of the glutathione metabolism during cold acclimation in wheat varieties with different freezing tolerance. J Agronomy Crop Science 208:65–75. https://doi.org/10.1111/jac.12566

Asghar MA, Kulman K, Szalai G, et al (2023) Effect of ascorbate and hydrogen peroxide on hormone and metabolite levels during post‐germination growth in wheat. Physiologia Plantarum. https://doi.org/10.1111/ppl.13887

Reviewer 2 Report

The paper "Biochar-mediated control of metabolites and other physiological responses in water-stressed Leptocohloa fusca" is a topical study that adds to the literature, especially in the way it statistically analyzes the data obtained.

The only comments I can give to the authors are the following:

1. if figures 3 and 4 can be redone in terms of notations (if they are made in originlab please bold the notations to see more clearly)

2. within the paper there are two notations of figure 4 - page 12, line 317 and page 14, line 371. I assume the last one (page 14) is figure 5

Author Response

Response to Reviewer 2

Firstly, the authors wish to Thank for your valuable comments in improving the manuscript.

  1. if figures 3 and 4 can be redone in terms of notations (if they are made in originlab please bold the notations to see more clearly)

Response: The heatmaps were prepared with the MultiExperiment Viewer software and we have highlighted all the titles in the figures as much as we could.  By following this scheme of heatmaps, we published multiple papers in the well reputed journals (Asghar et al. 2022b, a, 2023). 

  1. within the paper there are two notations of figure 4 - page 12, line 317 and page 14, line 371. I assume the last one (page 14) is figure 5

Response: The correction has been made in the figure legends throughout the MS.

References

Asghar MA, Balogh E, Ahres M, et al (2022a) Ascorbate and Hydrogen Peroxide Modify Metabolite Profile of Wheat Differently. J Plant Growth Regul. https://doi.org/10.1007/s00344-022-10793-0

Asghar MA, Balogh E, Szalai G, et al (2022b) Differences in the light‐dependent changes of the glutathione metabolism during cold acclimation in wheat varieties with different freezing tolerance. J Agronomy Crop Science 208:65–75. https://doi.org/10.1111/jac.12566

Asghar MA, Kulman K, Szalai G, et al (2023) Effect of ascorbate and hydrogen peroxide on hormone and metabolite levels during post‐germination growth in wheat. Physiologia Plantarum. https://doi.org/10.1111/ppl.13887

Reviewer 3 Report

This manuscript presents good research work with a lot of experimental and statistical data. Only a few minor concerns that should be corrected before final publication of this work.

Page 2, line 68: It says: "These reactive molecule", and should read "These reactive molecules".

The term "specie" appears several times on page 2. It would be better to replace it with the term "species".

Page 3, lines 121-122: please specify the difference between BC-1 and BC-2.

Page 5, line 215: please specify the SPPS version used.

Pages 6, 8 and 11: please do not leave a page almost blank, adjust the size of the graph to fit in the blanks on these pages.

Page 7: figure legend 1: please specify the meaning of the letters above the columns.

Page 9: Figure legend 2: Please specify the meaning of the letters above the columns.

The writing of the manuscript is a bit cumbersome because it includes a lot of data. Perhaps it could be presented in a more didactic way.

Author Response

Response to Reviewer 3

  1. Page 2, line 68: It says: "These reactive molecule", and should read "These reactive molecules".

Response: The suggested change has been made in the revised MS (page 2).

  1. The term "specie" appears several times on page 2. It would be better to replace it with the term "species".

Response: The requested corrections have been made throughout the page 2.

  1. Page 3, lines 121-122: please specify the difference between BC-1 and BC-2.

Response: BC-1 and BC-2 were the two different levels of biochar (15 and 30 mg/kg respectively), mixed in garden soil. The additional information has been added (page 3) to the revised MS.

  1. Page 5, line 215: please specify the SPPS version used.

Response: The necessary information regarding SPSS version has been added (page 5) to the revised MS.

  1. Pages 6, 8 and 11: please do not leave a page almost blank, adjust the size of the graph to fit in the blanks on these pages.

Response: We have adjusted the size of the figures to justify the space.

  1. Page 7: figure legend 1: please specify the meaning of the letters above the columns.

Response: These are the letters of significance among the applied treatments for roots and shoots separately i.e. the treatments which were applied were compared in the roots separately and shoots separately. In addition one-way ANOVA was applied to compare the treatments.   

  1. Page 9: Figure legend 2: Please specify the meaning of the letters above the columns.

Response: These are the letters of significance among the applied treatments for roots and shoots separately i.e. the treatments which were applied were compared in the roots separately and shoots separately. In addition one-way ANOVA was applied to compare the treatments.   

Reviewer 4 Report

General features: The general content of the article is adequate and complete. The number of authors is very high (in relation to the work carried out). The language is correct. The writing is adequate and easy to understand. The structure of the text is correct. 2. Minor revisions:  Need to harmonize the style of graphics  It is necessary to update the bibliography including the citations that come from the current year 2023  The article is dated 2022. It must be updated to 2023

Author Response

Response to Reviewer 4

  1. Need to harmonize the style of graphics.

Response: The graph no. 1 and 2 are harmonized together and 3, 4, 5, and 6 are harmonized together based on the similarities.

  1. It is necessary to update the bibliography including the citations that come from the current year 2023. The article is dated 2022. It must be updated to 2023.

Response: We have updated the bibliography with the latest citations as much as we could.

Round 2

Reviewer 1 Report

Thank you very much for considering my remarks & comments valuable!